# Medical Comorbidities in *MECP2* Duplication Syndrome: Results from the International *MECP2* Duplication Database

**DOI:** 10.3390/children9050633

**Published:** 2022-04-28

**Authors:** Daniel Ta, Jenny Downs, Gareth Baynam, Andrew Wilson, Peter Richmond, Helen Leonard

**Affiliations:** 1Telethon Kids Institute, University of Western Australia, Perth, WA 6009, Australia; jenny.downs@telethonkids.org.au (J.D.); gareth.baynam@health.wa.gov.au (G.B.); andrew.wilson@health.wa.gov.au (A.W.); peter.richmond@uwa.edu.au (P.R.); helen.leonard@telethonkids.org.au (H.L.); 2Curtin School of Allied Health, Curtin University, Perth, WA 6102, Australia; 3WA Department of Health, Genetic Services of Western Australia, Subiaco, WA 6008, Australia; 4Western Australian Register of Developmental Anomalies, King Edward Memorial Hospital, Subiaco, WA 6904, Australia; 5North Entrance, Perth Children’s Hospital, 15 Hospital Ave., Nedlands, WA 6009, Australia; 6Discipline of Paediatrics, School of Medicine, University of Western Australia, Perth, WA 6009, Australia

**Keywords:** *MECP2* duplication syndrome, neurodevelopmental disorder, intellectual disability, epilepsy, recurrent infections

## Abstract

Since the discovery of *MECP2* duplication syndrome (MDS) in 1999, efforts to characterise this disorder have been limited by a lack of large datasets, with small case series often favouring the reporting of certain conditions over others. This study is the largest to date, featuring 134 males and 20 females, ascertained from the international *MECP2* Duplication Database (MDBase). We report a higher frequency of pneumonia, bronchitis, bronchiolitis, gastroesophageal reflux and slow gut motility in males compared to females. We further examine the prevalence of other medical comorbidities such as epilepsy, gastrointestinal problems, feeding difficulties, scoliosis, bone fractures, sleep apnoea, autonomic disturbance and decreased pain sensitivity. A novel feature of urinary retention is reported and requires further investigation. Further research is required to understand the developmental trajectory of this disorder and to examine the context of these medical comorbidities in a quality of life framework.

## 1. Introduction

MECP2 duplication syndrome (MDS; OMIM 300260) is a rare, X-linked, neurodevelopmental disorder caused by a duplication of the methyl-CpG-binding protein 2 (MECP2) gene, with an estimated birth prevalence of 1/150,000 liveborn males [1]. In contrast, loss-of-function mutations of MECP2 lead to Rett syndrome (RTT; OMIM 312750), a similar disorder with phenotypic overlap.

Commonly reported features of MDS include intellectual disability, seizures, lower-respiratory-tract infections (LRTIs), gastrointestinal problems, symptoms of autism, some dysmorphic features, sleep disturbances and abnormal neuroradiological findings. Previous studies have provided limited attention to certain clinical areas of MDS including other infections (e.g., pharyngitis, tonsilitis, otitis and urinary tract infections [UTIs]) cardiovascular defects, urogenital abnormalities beyond cryptorchidism, behaviour or mood disturbances (other than autism) and autonomic dysfunction [2,3,4].

Since the first description of MDS in 1999 [5], the largest case series has featured 59 males [3]. The current literature, much of which comprises small case series, reflects the lack of a comprehensive phenotypic description. To address this gap, the international MECP2 Duplication Database (MDBase) was established in 2020 to ascertain a large sample size and collect comprehensive health-related data from caregivers of individuals with MDS. The current study has utilised this database to examine the prevalence and onset of MDS medical comorbidities to expand the phenotypic knowledge base.

## 2. Materials and Methods

### 2.1. Data Source

The international *MECP2* Duplication Database (MDBase) collects parent-reported, health-related data internationally on individuals with MDS. An MDS-specific family questionnaire, formulated to comprehensively capture clinical features and natural history, was developed following consultation with a consumer reference group of family stakeholders. The questionnaire collects information on medical comorbidities including cardiopulmonary health (relating to conditions involving the heart, lungs and autonomic nervous system), epilepsy (relating to onset of seizures and treatment), gastrointestinal (GI) health (relating to GI problems, feeding and surgical interventions), ear, nose, and throat (ENT) health (relating to otitis media, sinusitis, tonsillitis and pharyngitis and its treatment), musculoskeletal health (relating to onset of spinal curvature and bone fractures), kidney/urinary tract health (relating to UTIs, urinary retention and kidney problems), pain sensitivity (relating to increased or decreased reaction to pain) and sleeping characteristics. Other sections of the questionnaire queried developmental milestones and functional abilities, behaviours (relating to hand stereotypies, bruxism, developmental behaviour and RTT-like behaviour), puberty, medications, hospital admissions, day activities and school options, family structure and demographics and the quality of life of individuals with MDS, their family and main caregiver, although these are not analysed in the current paper.

Data on individuals with MDS had previously been included in the International Rett Syndrome Database (InterRett) [6]. As such, families of these individuals were invited to participate in this new database. Further ascertainment occurred through the following sources: social media advertising on family support groups on Facebook, advertising via our study website (https://rett.telethonkids.org.au/about/mecp2-duplication-syndrome/, accessed on 30 July 2021), word of mouth at the 2020 *MECP2* Duplication Syndrome Conference (Figure 1). Phone, videoconferencing and email contact was established with newly ascertained families or re-established with families from InterRett. They were then invited to complete the online questionnaire and provide genetic reports where possible.

Ethics approval was obtained from the University of Western Australia Human Research Ethics Committee (2019/RA/4/20/5929).

### 2.2. Data Analysis

Descriptive statistics were reported as the median and range values for continuous variables or proportions and percentages for categorical variables. Frequencies of different comorbidities were compared by gender using Fisher’s exact test. Time-to-event analysis was performed to estimate the median age of developing seizures and spinal curvature. All statistical analysis were performed using STATA software.

## 3. Results

### 3.1. Study Population

As of the census date (1 November 2021) for this study, questionnaires had been administered to families of 205 individuals with MDS. The main source of ascertainment was Facebook (*n* = 119 [58%]), followed by InterRett (*n* = 40 [20%]), the 2020 *MECP2* Duplication Syndrome Family Conference (*n* = 15 [7%]), our study website (*n* = 13 [6%]) and sharing of our study on MDS-related websites and by word of mouth (*n* = 18 [9%]). Fully completed questionnaires were provided for 138 (67%) and partially completed for a further 16 (8%). Data were available for 134 males [87%] and 20 females [13%]; Table 1, including two male sibling pairs and 15 individuals (13 male, 2 female) who died prior to the census date. Most individuals were from North America (*n* = 76 [49%]), followed by Europe (*n* = 51 [33%]), Oceania (*n* = 21 [14%]), Asia (*n* = 5 [3%]) and South America (*n* = 1 [1%]). The median age at data completion was 8.76 years for males (range = 0.94–51.59 years) and 10.17 years for females (range = 4.88–31.36 years).

### 3.2. Cardiopulmonary Health

Congenital heart disease (CHD) was reported in just under one-fifth of individuals (27/146 [18%]), with some individuals having more than one heart defect (Table 2. The most common defect was either an atrial or ventricular septal defect (*n* = 17) followed by patent ductus arteriosus (*n* = 3), pulmonary vein stenosis (*n* = 2), coronary artery anomaly (*n* = 2), and one each for bicuspid aortic valve, accessory tricuspid valve tissue, patent foramen ovale, ventricular hypertrophy, pulmonary vein anomaly and Brugada syndrome. Pulmonary hypertension was reported in three of 147 (2%) individuals and was the cause of premature death in one male individual at the age of 11 months.

Pneumonia was the most common respiratory infection reported (Figure 2), with between three-fifths and three-quarters of individuals being affected across all age groups from birth to over 20 years of age. Episodes of bronchitis decreased through childhood and adolescence but then increased in those over 20 years of age. Bronchiolitis had been experienced by more than half of individuals under the age of 5 years (77/138 [56%]; Table 2). The proportions of individuals that had experienced pneumonia, bronchitis and bronchiolitis were all significantly higher in males than in females (*p* = 0.017, 0.013 and *p* = 0.044, respectively).

The prevalence of asthma in this population remained consistent, affecting approximately one-fifth to one-quarter of individuals in all age groups. Croup was less frequent—reported to occur in one-quarter (31/133 [23%]) of individuals less than 5 years of age and less than one-fifth of individuals older than 5 years of age (between 6% and 18% of individuals in five-year epochs).

More than half (77/135 [57%]) of individuals had received antibiotics for a respiratory infection in the past 12 months: most (*n* = 51) received a short-term course of antibiotic (each lasting less than six weeks), while some (*n* = 9) received a long-term/preventative course of antibiotics (each lasting more than six weeks) or a combination of both short-term and preventative antibiotics (*n* = 17). Prior to the past year, 41/139 (29%) individuals had received a prolonged course of prophylactic antibiotics for a respiratory infection, with the main prophylactics being azithromycin, amoxicillin and trimethoprim.

Most individuals (118/146 [81%]) had been hospitalised at least once due to a respiratory problem. For those who had ever been hospitalised for a respiratory problem, the median number of estimated hospitalisations increased slightly over successive and specific age epochs for both males and females (Table 3).

Over a half (81/147 [55%] of individuals had problems with aspiration either occasionally or frequently, with caregivers reporting that almost one-quarter (33/146 [23%]) had aspirated fluid or food into their lungs in the past 12 months. More than a half (78/146 [53%]) were reported to be unable to effectively cough up chest secretions. Special equipment and/or therapies were used by 77/147 [52%] of individuals (70/127 males, 7/20 females) to assist with management of respiratory problems. Airway clearance devices and/or techniques including suction machines, cough assist machines, positive expiratory pressure (PEP) masks, vest therapy, chest physiotherapy and postural drainage were utilised by 57/147 (39%; 52/127 males, 5/20 females). One-fifth (28/147 [19%]; 24/127 males, 4/20 females) used oxygen therapy or an oxygen concentrator. Similarly, 21/147 (14%; 19/127 males, 2/20 females) individuals required bilevel positive airway pressure (BIPAP), continuous positive airway pressure (CPAP) or automatic positive airway pressure (APAP) machines to aid in breathing.

More than four-fifths (119/146 [81%]) of individuals were reported to have some evidence of autonomic disturbance, either previously, currently, or both. Over two-thirds (101/145 [70%]) had cold peripheries (Table 2), while over a half (83/146 [57%]) had problems regulating body temperature and 38% (55/144) had experienced episodes of breath holding. Few individuals experienced hyperventilation (18/144 [13%]).

### 3.3. Epilepsy

Seizures were reported in over 60% (90/148) of individuals (Table 2), with a median onset of 8 years in males and 9.8 years in females estimated using time-to-event analysis (Figure 3). Reported seizure types (Figure 4) were tonic (60/87 [70%]), absence (56/87 [64%]), atonic (53/87 [61%]), clonic (53/87 [61%]), myoclonic (53/87 [61%]), tonic-clonic (42/87 [48%]), focal–simple (29/87 [33%]), focal–complex (24/87 [28%]), gelastic (23/87 [26%]) and infantile spasms (18/87 [21%]). Where data were available, 71/87 (82%) individuals experienced more than one type of seizure. One-quarter (22/89 [25%]) were also diagnosed with Lennox–Gastaut syndrome (LGS).

Caregivers reported the most common seizure trigger as infection (35/56 [63%]) followed by fever (30/56 [54%]), arousal from sleep (29/56 [52%]), sleep deprivation (22/56 [39%]), emotional upset (15/56 [27%]), feeding (14/56 [25%]), sudden noise (13/56 [23%]) and facial tactile stimuli (3/56 [5%]).

The most frequently used anti-seizure medications (ASMs) or medicinal cannabis preparations used included levetiracetam (55/74 [74%]), valproate/valproic acid (VPA; 43/74 [58%]), clobazam (31/74 [42%]), lamotrigine (25/74 [34%]) and epidiolex/CBD (14/74 [19%]; Appendix A). A median of two ASMs were being used by each individual. A single monotherapy or polytherapy regime that successfully controlled seizures was not identified but the highest proportion reported valproate to be effective (23/43 [53%]).

More than two-thirds (60/87 [69%]) had treatment-refractory seizures. Approximately one-third (27/87 [31%]) had been diagnosed with status epilepticus, with a median of 2–3 (range: 1–50) episodes per person. A half (46/86 [53%]) had on occasions needed rescue medications including diazepam, midazolam and lorazepam—and less commonly, clonazepam, propofol, phenytoin and paraldehyde.

Eleven individuals had trialled a ketogenic diet to control seizures beginning at a median age of 10 years (range: 1.8–16.5 years). Improvements in seizure control were reported for nine including by frequency (*n* = 8), duration (*n* = 5) and severity (*n* = 5). For one individual, no changes were observed; and for another, there were no data available. The median time for these improvements to occur was three weeks (range: 1–12 weeks). Two individuals who experienced benefits from the ketogenic diet had to cease the diet because of the development of pancreatitis in one and constipation in the other.

Six individuals had a vagus nerve stimulator (VNS) inserted to control seizures at a median age of 13.9 years (range: 7–33 years). Improvement in seizure control was reported for four including by frequency (*n* = 2), duration (*n* = 2) and severity (*n* = 2). For two individuals, there was no change in seizure control. Where data were available, improvements were observed at 2 weeks, 4 weeks and 6 months for three individuals, respectively, following insertion of VNS. One individual had had a corpus callosotomy without any apparent benefit to seizures.

### 3.4. Gastrointestinal Health

Almost all (130/142 [92%]) individuals had experienced a gastrointestinal disorder (Table 2). The most commonly reported problem was constipation (126/135 [93%]), followed by gastroesophageal reflux in four-fifths (107/132 [81%]), slow gut motility in under a half (58/123 [47%]), air swallowing in one-quarter (35/128 [27%]) and intestinal pseudo-obstruction in one-fifth (23/122 [19%]). Males experienced a significantly higher proportion of gastroesophageal reflux and slow gut motility than females (*p* = 0.020 and *p* = 0.031, respectively).

Feeding difficulties were reported in two-thirds (96/145 [66%]). A little under two-thirds (89/138 [64%]) took all food and drinks orally. For others, all food and drink were provided either via a gastrostomy button (41/138 [30%]), gastrojejunostomy button (6/138 [4%]) or a nasogastric tube (2/138 [1%]). Of those with a gastrostomy button, some (*n* = 22) also received nutritional supplements, liquids and/or medications through the button—similarly for those with a gastrojejunostomy button (*n* = 3) and nasogastric tube (*n* = 2).

The median age of insertion of a gastrostomy, gastrojejunostomy or nasogastric tube was 5.9 years (range: 1 month–19 years). Reason/s for the insertion of a feeding tube reported by caregivers included feeding difficulties (46/52 [88%]), to reduce respiratory infections (33/52 [63%]), to give medication (32/52 [62%]), to increase health and wellbeing (32/52 [62%]), to gain weight (29/52 [56%]), individual being too tired to eat (16/52 [31%]), abdominal bloating (3/52 [6%]) and to facilitate a ketogenic diet (1/52 [2%]).

Twelve individuals underwent a Nissen fundoplication at a median age of 1.75 years (range: 2 months–10 years) to treat gastroesophageal reflux disease (GERD)—all reporting positive outcomes.

### 3.5. Ear, Nose and Throat (ENT) Health

Ear infections were frequently reported, with over half (88/145 [61%]) experiencing an ear infection during their lifetime (Table 2). One-third (26/88 [30%]) had experienced one to two episodes, one-fifth (16/88 [18%]) three to four episodes and a half (46/88 [52%]) more than five episodes. Of these, 32/86 (37%) had grommets inserted.

Over one-third (50/140 [36%]) had experienced sinusitis and over half (88/140 [63%]) one or more episodes of pharyngitis/tonsillitis. Of those who experienced pharyngitis/tonsillitis, 40/84 (48%) had had a tonsillectomy and 8/84 (10%) also an adenoidectomy. A higher proportion of females (89%) than males (59%) had experienced pharyngitis/tonsillitis (*p* = 0.011).

### 3.6. Urogenital Problems

Over one-third (55/143 [38%]) of individuals had experienced at least one urinary tract infection (Table 2), with Escherichia coli and Pseudomonas spp. being the most commonly reported causative bacteria where data were available. More females (58%) than males (35%) had experienced urinary tract infections (*p* = 0.055).

Episodes of urinary retention were reported by caregivers in 41/142 (29%) individuals (Table 2), whereby urination often only occurred once or twice a day. Of those individuals who had experienced urinary retention, 14/40 (35%) had been observed to retain their urine for longer than a day. Although not comprehensively captured, three individuals received catheterisation for urinary retention, following which a reduction in UTIs was reported for two of the three. Five individuals were reported to have experienced renal calculi and two to have renal cysts identified.

### 3.7. Musculoskeletal Problems

Scoliosis was reported to have affected one-quarter (33/143 [23%]) of individuals (Table 2), two of whom also had kyphosis. Using time-to-event analysis, there was an estimated 25% likelihood of developing scoliosis by 11 years in males and 7.4 years in females (Figure 5). Nine had been treated with bracing from a median age of 9.6 years (range: 5–12 years) for a median duration of 1.5 years (range: 6 months–3.8 years). Two individuals had received surgical treatment for scoliosis, with beneficial results.

One-third (46/142 [32%]) of individuals had experienced a bone fracture, with most incidents being seizure or accident/fall related. Some individuals (19/143 [13%]) had been diagnosed with osteoporosis or osteopenia. In just under one-third (41/140 [29%]), additional muscle and/or orthopaedic conditions such as hip dysplasia and/or dislocation (*n* = 18), muscle contractures (*n* = 17) and septic arthritis (*n* = 2) were reported.

### 3.8. Abnormal Pain Sensation

More than half (85/143 [59%]) were reported to have abnormal pain sensitivity (Table 2). Primarily, caregivers reported a decreased sensitivity to pain (79/143 [55%]). Examples of pain insensitivity include a delayed or no reaction to needles (i.e., vaccination and canulation) or injuries such as falls, burns or broken bones. Less commonly reported was increased pain sensitivity (13/143 [9%]), with examples including sensitivity and pain to touch or temperature (i.e., brushing hair and warm weather) and being particularly upset over slight pain.

### 3.9. Sleep

Sleep apnoea was diagnosed in 61/143 (43%; Table 2) individuals, at a median age of 2.5 years (range: 3 months–25 years). Two-thirds (40/59 [68%]) of those with sleep apnoea had been surgically treated: adenotonsillectomy (*n* = 31), adenoidectomy (*n* = 7; two also with a supraglottoplasty) and tonsillectomy (*n* = 2). Benefits from surgery were reported for three-quarters (19/25 [76%]) of individuals, while one caregiver reported no benefit from adenotonsillectomy and five were awaiting results of another sleep study or were unsure. Breathing assistance was provided via use of CPAP (*n* = 15) and BIPAP (*n* = 6) machines for sleep breathing disorders. Under one-third (*n* = 18) also received oxygen therapy.

## 4. Discussion

This largest study to date provides in-depth phenotyping of 134 males and 20 females with MDS. We confirm the higher prevalence of pneumonia, bronchiolitis, bronchitis, gastroesophageal reflux and slow gut motility in males than females. We also confirm the presence of seizures, gastrointestinal problems including constipation and feeding difficulties, scoliosis, fractures, sleep apnoea, autonomic disturbance, decreased pain sensitivity and urinary retention in both males and females affected by this disorder [3,7]. We expand upon the infectious diseases profile of individuals with MDS by characterising the frequency of ENT infections and UTIs in addition to lung infections. This is also the largest study to characterise the presence of congenital heart disease in individuals with MDS and also to note the frequent occurrence (29%) of urinary retention.

A previous clinical sample reported a milder phenotype in females with MDS [8], such that they (*n* = 5) scored lower than males (*n* = 43) on a RTT Clinical Severity Scale (CSS) [7], while our previous case series found that females had better motor ability [9]. However, this current much larger study shows that whilst males may have a greater burden of respiratory infections and gastrointestinal disorders, females can still display phenotypic penetrance and some manifest the same medical comorbidities and similar medical burden as males. Our current study does not compare the level of functional abilities, development or behaviour between males and females—an area that will be explored in the future from this current, larger dataset.

Congenital heart disease is a leading cause of birth defects, affecting an estimated 0.8% to 1.2% live births worldwide and is associated with a slight increase in mortality [10,11]. We found its prevalence in our MDS study population to be much greater, at 18%, but lower when compared to other common genetic syndromes such as Down syndrome (40% to 50%) and Turner syndrome (25% to 45%) [12]. Nevertheless, this may still be of clinical significance as the most common defect reported was either an atrial or ventricular septal defect while pulmonary hypertension has now been the reported cause of death in four cases in the literature, three of whom were under 2 years of age [1,3,13].

This study also provides a clearer understanding of the frequency of infections in MDS and differentiating between pneumonia and other respiratory infections. Consistent with previous studies, pneumonia was a major health issue and was the key LRTI in our study population [3,14]. Our study is also the first to extensively characterise the trajectory of episodes of pneumonia and bronchitis in five-year epochs and thus provide information about the natural history of respiratory infections. Unsurprisingly, males were disproportionally affected to females, highlighting the burden of LRTIs in the MDS male phenotype. It has previously been shown that some individuals with MDS have an underlying immunological problem (specifically IgA/IgG2 deficiency or functional antibody deficiency against pneumococcal vaccination) postulated to be a causal factor in early mortality [14]. It is also possible that the presence of gastroesophageal reflux leading to aspiration pneumonia may be contributing to the prevalence of respiratory infections, as was reported in a previous study [5]. The bimodal pattern of bronchitis (Figure 2) may indicate progressive chronic suppurative lung disease in adult life which requires further investigation and may influence management. Approximately one-quarter (23%) of individuals with MDS appear to have experienced croup, which only affects approximately 3% of typically developing children and usually between the ages of six months and three years [15], a finding which may suggest possible underlying upper-airway abnormality in individuals with MDS particularly given its persistence into adolescence. Additionally, other URTIs such as pharyngitis/tonsillitis, otitis and sinusitis were reported in 63%, 61% and 36% of individuals, respectively, and UTIs in 38% of individuals. We are providing a clearer picture of the propensity to infections, possibly related to the immunological deficiency, that individuals with MDS may face [14].

Whilst the general population prevalence of asthma in children is difficult to estimate because of definitional issues [16] the 2017–2018 Australian Bureau of Statistics (ABS) National Health Survey (NHS) estimates that approximately 11.2% of Australians have asthma based on self-reported data [17]. In comparison, approximately one-quarter of individuals with MDS in our study were reported to have experienced asthma, possibly reflecting misdiagnosis/overdiagnosis as ongoing bronchial hyperresponsiveness is likely a reflection of aspiration lung disease causing an asthma-like phenotype as is reported in cerebral palsy [18].

Reports of autonomic disturbances in previous studies have been scant [1,19,20,21,22], with the most commonly reported issue being vasomotor problems (e.g., moist and red/cold hands and/or feet, and livedo of limbs) in 58 of 100 individuals from three studies [3,7,23]. Expanding on these findings, our study found that over 70% of individuals were reported to experience cold peripheries and 57% problems regulating body temperature. Breath-holding spells have been hypothesised to be associated with autonomic nervous system dysregulation [24,25]. Whilst breath holding or hyperventilation has only rarely been identified in individuals with MDS previously [19,20,26] our study found that 38% and 13% of individuals were reported to experience episodes of breath holding and hyperventilation, respectively. Autonomic nervous system dysregulation is a known feature of Rett syndrome, where two-thirds of individuals experience breath-holding and just under a half hyperventilation [27]. It is plausible that autonomic dysfunction may also be a feature of MDS.

Our large sample size and the use of time-to-event analysis account for the age of the individuals at data collection to provide a better estimate of age at seizure onset [28], with a median age of 8 years and 9.8 years for males and females, respectively. Other studies involving clinic samples which have not used time-to-event analysis [28] and thus not accounted for the age of their participants have recorded lower age at onset [3,23,29,30]. The two most frequently used anti-seizure medications (ASMs), levetiracetam and VPA, in our study were similarly reported in a US study by Marafi et al. [29]. Interestingly, the use of VPA in our study was associated with a higher proportion of individuals reporting effectiveness in controlling seizures and fewer side effects. The effectiveness of VPA has been recently documented in a case of MDS in which the individual initially trialled the ASM and remained seizure free [31]. Valproate was initially discontinued due to thrombocytopenia and hyperbilirubinemia. Seizures were then refractory to multiple other ASMs. Subsequently, the patient was re-challenged with VPA with close monitoring of platelet and bilirubin levels, resulting in a successful decrease in drop attacks from 20 to 30 times per day to 1–2 times daily. Together, this evidence might suggest that VPA is an effective first line ASM for generalised seizures for individuals with MDS.

An interesting finding in our study was the reporting of urinary retention in 41/142 (29%) of individuals, including retention for greater than 24 h in some. Urine retention has been reported as a rare side effect of ASMs such as carbamazepine [32], levetiracetam [33], felbamate [34], clobazam [35], and phenytoin [36]. Of the 41 individuals experiencing urinary retention/incontinence, 33 individuals received either one or more of such ASMs, no data were available for six individuals and only two individuals did not receive any ASMs. The episodes of UTIs experienced by over one-third of individuals could also be associated with urinary retention. The authors propose that urinary retention may be at the centre of a complex intersection of factors related to ASMs and UTIs that warrants further investigation.

The prevalence of scoliosis in previous studies was higher—for example, in the French cohort, 23/43 males (53%) had scoliosis and/or kyphosis compared to 33/143 (23%) in our study. It should be noted that the median age of the French cohort was 12 years (range: 2–48 years), older than that of our cohort, which was 9.16 years (range: 0.94–51.59 years), which may explain the difference in prevalence. Interestingly, we noted approximately one-third (32%) of individuals had experienced bone fractures being seizure or accident/fall related. Whilst 19/143 (13%) of individuals were diagnosed with low bone density, it should be noted that the use of valproate for epilepsy in RTT may also increase the risk of bone fractures [37] and could have implications for the use of this medication in MDS.

Decreased pain sensitivity reported in just over half (55%) of individuals was less frequent compared to previous studies [3,9,38,39,40]. Pain insensitivity is also a common issue in RTT, as a study using Australian and international data reported that 317/497 (64%) individuals with RTT had decreased sensitivity to pain [41]. Similarly, families of individuals with RTT reported increased tolerance of falls, trauma and procedures such as blood drawing and injections [41]. An animal model of MDS has shown that overexpression of MeCP2 provides an analgesic role in acute mechanical pain and thermal pain transduction [42]. The understanding of impaired nociception in MDS has clinical implications for the recognition of injuries. On the other hand, increased pain sensitivity was less commonly reported in 9% of individuals in this study. Similarly, increased sensation to pain was also less commonly reported in the Rett syndrome study in 50/497 (10%) individuals [41].

The key limitation of this study was the selection bias of individuals with MDS in families from primarily English-speaking backgrounds as the questionnaire was not translated to other languages due to lack of funding. As a result, most families were recruited from the USA, the UK or Australia. Furthermore, an unknown proportion of individuals reported in this study have previously been reported in the literature. Health data ascertained in this study have also been parent reported and might be subject to recall error. To limit recall error, we asked caregivers to refer to medical records where available. Other domains that require further characterisation within this dataset include the developmental domains and behavioural patterns of hand stereotypies, bruxism, sleep and autistic behaviours, each associated with RTT and thus will provide a clearer delineation between the behavioural phenotype of these two similar disorders [43]. However, further information on the developmental trajectory of this disorder and how it varies between males and females including gross motor, communication and hand function milestones and the possible regression of such skills are required to better understand disease progression in MDS and to similarly differentiate from the well-established patterns of developmental regression in RTT [43].

In summary, this study provides a comprehensive evaluation of medical comorbidities in MDS and a valuable baseline for comparison with RTT. We highlight the burden of respiratory infections and gastrointestinal disorders in the male phenotype and demonstrate that phenotypic penetrance in the female phenotype presents a similar disease burden to males. The size of our community sample has provided a more accurate understanding of the likelihood of developing seizures and spinal curvature in MDS and an expanded phenotype that may be important for clinical care. The high rates of medical comorbidities require further work to understand the impact on quality of life as well as management.

## Figures and Tables

**Figure 1 children-09-00633-f001:**
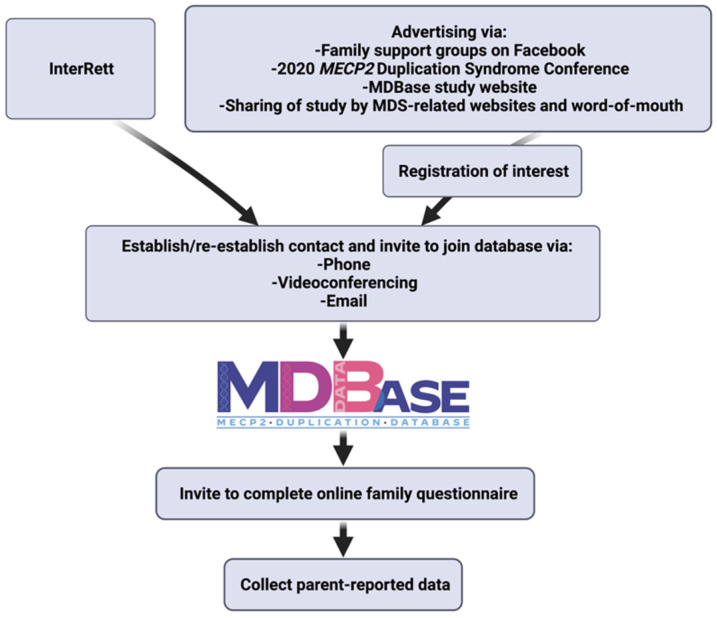
Method of data collection (Created with BioRender.com, accessed on 18 March 2022).

**Figure 2 children-09-00633-f002:**
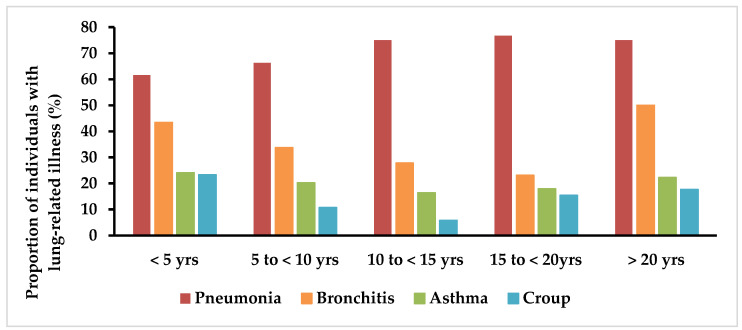
Frequency distribution of pneumonia, bronchitis, asthma and croup being sometimes or constantly a problem during 5 year epochs. Note: Sample size decreases with each increasing 5 year epoch.

**Figure 3 children-09-00633-f003:**
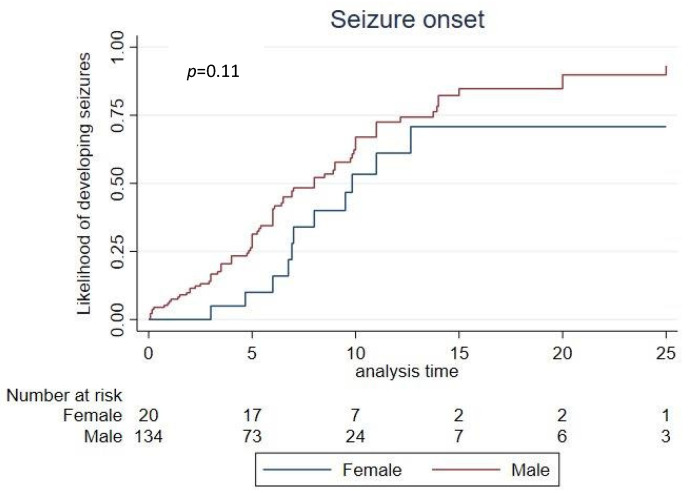
Time-to-event estimates of developing seizures.

**Figure 4 children-09-00633-f004:**
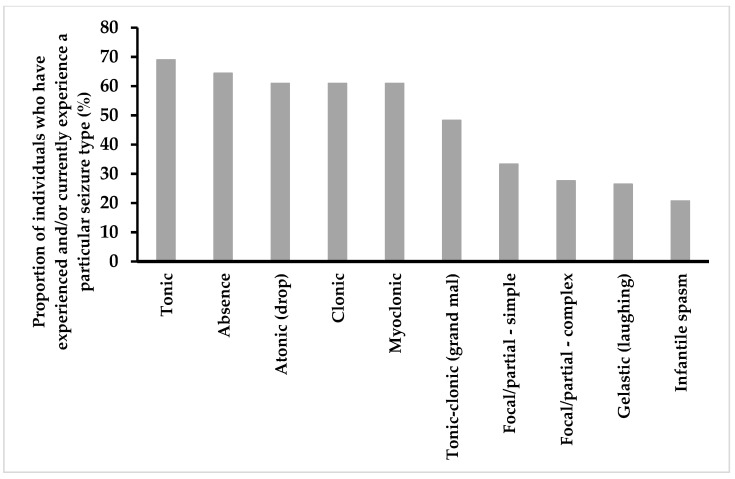
Proportion of individuals who have experienced different seizure types.

**Figure 5 children-09-00633-f005:**
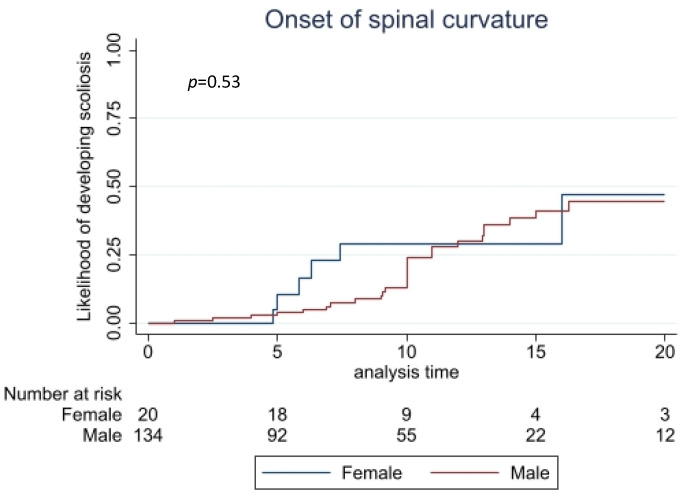
Time-to-event estimates of developing scoliosis.

**Table 1 children-09-00633-t001:** Demographics of male and female individuals with MDS partaking in the MDBase study.

	Males	Females	Total
**Median age** **(range)**	8.76 years (0.94–51.59 years)	10.76 years (4.88–31.36 years)	9.16 years (0.94–51.59 years)
	** *N* **	** *N* **	** *N* **
**0–<5 years**	40	1	12
**5–<10 years**	34	8	42
**10–<15 years**	27	7	34
**15–<20 years**	13	1	14
**20+ years**	20	3	23
**All age groups (%)**	134 (87%)	20 (13%)	154 (100%)
**Continent**		***N* (%)**	
North America ^1^		76 (49%)	
Europe ^2^		51 (33%)	
Oceania ^3^		21 (14%)	
Asia ^4^		5 (3%)	
South America ^5^		1 (1%)	

^1^ The USA and Canada; ^2^ the UK, Italy, Netherlands, Spain, Poland, Germany, Norway, Denmark, Belgium, France, Finland, Sweden, Switzerland, Andorra, Hungary, Romania and Russia; ^3^ Australia and New Zealand; ^4^ Japan, Taiwan, India and United Arab Emirates; ^5^ Brazil.

**Table 2 children-09-00633-t002:** Medical comorbidities in individuals with MDS.

Feature	Males	Females	Total	*p*-Value *
	*n*/*N* (%)	*n*/*N* (%)	*n*/*N* (%)	
*Respiratory problems*				
Pneumonia	89/122 (73%)	8/18 (44%)	97/140 (70%)	**0.017**
Bronchitis	60/118 (51%)	4/19 (21%)	64/137 (47%)	**0.013**
Bronchiolitis	76/120 (63%)	7/18 (39%)	83/138 (60%)	**0.044**
Asthma	31/119 (26%)	3/18 (17%)	34/137 (25%)	0.295
Croup	29/116 (25%)	2/17 (12%)	31/133 (23%)	0.187
Aspiration	72/127 (57%)	9/20 (45%)	81/147 (55%)	0.231
*Seizures/epilepsy*	78/128 (61%)	12/20 (60%)	90/148 (61%)	0.561
Treatment-refractory seizures	51/75 (68%)	9/12 (75%)	60/87 (69%)	0.453
Lennox–Gastaut syndrome	20/77 (26%)	2/12 (17%)	22/89 (25%)	0.386
*Gastrointestinal problems*	115/124 (93%)	15/18 (83%)	130/142 (92%)	0.180
Constipation	111/117 (95%)	15/18 (83%)	126/135 (93%)	0.100
Intestinal pseudo-obstruction	21/106 (20%)	2/16 (13%)	23/122 (19%)	0.381
Reflux	97/115 (84%)	10/17 (59%)	107/132 (81%)	**0.020**
Air swallowing	32/111 (29%)	3/17 (18%)	35/128 (27%)	0.258
Slow gut motility	54/106 (51%)	4/17 (24%)	58/123 (47%)	**0.031**
Gallbladder problems	6/103 (6%)	1/17 (6%)	7/120 (6%)	0.667
*Ear, nose and throat (ENT) infections*				
Otitis	76/126 (60%)	12/19 (63%)	88/145 (61%)	0.512
Sinusitis	45/122 (37%)	5/18 (28%)	50/140 (36%)	0.318
Pharyngitis/tonsillitis	72/122 (59%)	16/18 (89%)	88/140 (63%)	**0.011**
*Urogenital problems*				
Urinary tract infection	44/124 (35%)	11/19 (58%)	55/143 (38%)	**0.055**
Urinary retention	37/123 (30%)	4/19 (21%)	41/142 (29%)	0.304
Episodes lasting more than a day	13/36 (36%)	1/4 (25%)	14/40 (35%)	0.562
*Scoliosis*	27/124 (22%)	6/19 (32%)	33/143 (23%)	0.250
*Congenital heart disease (CHD)*	21/127 (17%)	6/20 (30%)	27/147 (18%)	0.130
*Autonomic problems*				
Cold peripheries	90/125 (72%)	11/20 (55%)	101/145 (70%)	0.103
Episodes of hyperventilation	16/124 (13%)	2/20 (10%)	18/144 (13%)	0.528
Episodes of breath holding	48/125 (38%)	7/19 (37%)	55/144 (38%)	0.555
Problems regulating body temperature	75/126 (60%)	8/20 (40%)	83/146 (57%)	0.082
*Abnormal pain sensation*	71/124 (57%)	14/19 (74%)	85/143 (59%)	0.133
Decreased pain sensitivity	67/124 (54%)	11/19 (58%)	78/143 (55%)	0.475
Increased pain sensitivity	5/124 (4%)	3/19 (16%)	8/143 (6%)	0.073
*Sleep apnoea*	54/123 (44%)	6/19 (32%)	60/142 (42%)	0.225

* Fischer’s exact test for testing the equality of two proportions.

**Table 3 children-09-00633-t003:** Proportion of individuals who have been hospitalised for a respiratory problem per age epoch and the median number of estimated hospitalisations.

Age Range	Males			Females			Total		
	*n*/*N* (%)	Med Times (Range)		*n*/*N* (%)	Med Times (Range)		*n*/*N* (%)	Med Times (Range)	
Birth to 12 months	56/78 (72%)	*N* = 56	2 (1–20)	5/11 (45%)	*N* = 5	2 (1–6.5)	61/89 (69%)	*N* = 61	2 (1–20)
12 to 24 months	65/75 (87%)	*N* = 65	2 (1–50)	10/12 (83%)	*N* = 10	2.5 (1–6.5)	75/87 (86%)	*N* = 75	2 (1–50)
2 to 5 years	55/64 (86%)	*N* = 55	3 (1–20)	9/11 (82%)	*N* = 9	4 (1–18)	64/75 (85%)	*N* = 64	3 (1–20)
5 to 10 years	44/51 (86%)	*N* = 44	2 (1–30)	5/10 (50%)	*N* = 5	4 (2–15)	49/61 (80%)	*N* = 49	3 (1–30)
10 to 15 years	32/35 (91%)	*N* = 32	2.25 (1–40)	3/4 (75%)	*N* = 3	6 (1–7)	35/39 (90%)	*N* = 35	2.5 (1–40)
15 to 20 years	15/20 (75%)	*N* = 15	5 (1–20)	2/2 (100%)	*N* = 2	8 (1–15)	17/22 (77%)	*N* = 17	5 (1–20)
Over 20 years	12/13 (92%)	*N* = 12	6.5 (1–28)	0/1 (0%)	N/A		12/14 (86%)	*N* = 12	6.5 (1–28)

N/A = not applicable.

## Data Availability

The data presented in this study are available on request from the corresponding author and subject to ethics approval.

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
