# Peer review of "Medical Comorbidities in MECP2 Duplication Syndrome: Results from the International MECP2 Duplication Database"

_children, 2022, doi:10.3390/children9050633_

Round 1

Reviewer 1 Report

In this manuscript, Ta et al. set out to assess clinical manifestation of MDS using the MDBase. This is the largest study to date and a few novel features of MDS are described. The study is certainly novel and timely. The manuscript is well written but a few typos and grammatical errors are scattered throughout. Overall I believe this is an important study that will provide much needed data for the field of MDS research.

There is a mixup of cross-references to certain tables throughout. E.g. Table 3 is referenced before Table 2. All cross-references need to be checked. 

L150ff - this data might be better presented in a Table in addition to the text. 

L172ff - This is a comment from a previous review and should be actioned and removed.

L306f - This needs a reference.

I feel that the authors over- and misuse the word 'ascertain'. To ease the flow I suggest replacing it  with recruited in L425.

I don't think the Appendix is referenced within the text.

Author Response

Thank you very much for your feedback on our work. We appreciate your comments and attention to detail. We do indeed have a few errors.

  • L178 refers to Table 2 which should be Table 3
  • Table 3 was missing a description, which has been added now (L304)
  • L172 was a comment from a previous review and has been removed

Furthermore:

  • We have accepted the suggestion for L150 and have included it in Table 3 now
  • L306 refers to the findings of this current paper and thus we do not believe it requires a reference
  • In L425, we have replaced the word 'ascertained' with 'recruited'
  • Appendix A is referenced in-text: please see L190

Please find all changes in the attached document. Thank you very much.

Reviewer 2 Report

In the submitted work Ta et al. present a largest to date featuring 134 males and 20 females with MDS. The authors provide a comprehensive evaluation of medical comorbidities in MDS and a valuable baseline for comparison with RTT. They highlight the burden of respiratory infections and gastrointestinal disorders in the male phenotype and demonstrate that phenotypic penetrance in the female phenotype presents a similar disease burden to males.

This work is certainly valuable to the medical staff handling patients with MDS and RTT. The manuscript is nicely written. I recommend to accept the manuscript as it stands.

Author Response

Thank you very much for your review of our article and kind words - we appreciate your time.

Reviewer 3 Report

very well written paper

Author Response

Thank you very much for your kind words and review of our paper.